# Estimating the costs of adolescent HIV care visits and an intervention to facilitate transition to adult care in Kenya

Enrique M. Saldarriaga[1,2]*, Kristin Beima-Sofie[2], Dalton Wamalwa[3,4], Cyrus Mugo[5,6], Irene Njuguna[2,5], Alvin Onyango[5], Grace John-Stewart[2,6,7], Monisha Sharma[2]*

1 The Comparative Health Outcomes, Policy, and Economics (CHOICE) Institute, University of Washington, Seattle, Washington, United States of America, 2 Department of Global Health, School of Public Health, University of Washington, Seattle, Washington, United States of America, 3 Department of Pediatrics & Child Health, University of Nairobi, Nairobi, Kenya, 4 Department of Pediatrics, Kenyatta National Hospital, Nairobi, Kenya, 5 Kenyatta National Hospital, Research and Programs, Nairobi, Kenya, 6 Department of Epidemiology, University of Washington, Seattle, Washington, United States of America, 7 Departments of Pediatrics and Medicine, University of Washington, Seattle, Washington, United States of America

* emsb@uw.edu (EMS); msharma1@uw.edu (MS)

## Abstract

### Introduction

Adolescents with HIV in sub-Saharan Africa face challenges transitioning to adult HIV care, which can affect long-term HIV care adherence and retention. An adolescent transition package (ATP) focused on transition tools can improve post-transition clinical outcomes, but its implementation costs are unknown.

### Methods

We estimated the average cost per patient of an HIV care visit and ATP provision to adolescents. Data was collected from 13 HIV clinics involved in a randomized clinical trial evaluating ATP in western Kenya. We conducted a micro-costing and activity-driven time estimation to assess costs from the provider perspective. We developed a flow-map, conducted staff interviews, and completed time and motion observation. ATP costs were estimated as the difference in average cost for an HIV care transition visit in the intervention compared to control facilities. We assessed uncertainty in costing estimates *via* Monte Carlo simulations.

### Results

The average cost of an adolescent HIV care visit was 29.8USD (95%CI 27.5, 33.4) in the standard of care arm and 32.9USD (95%CI 30.5, 36.8) in the ATP intervention arm, yielding an incremental cost of 3.1USD (95%CI 3.0, 3.4) for the ATP intervention. The majority of the intervention cost (2.8USD) was due ATP booklet discussion with the adolescent.

**Data Availability Statement:** All relevant data are within the manuscript and its Supporting Information files.

**Funding:** GJ-S and DW: This publication was made possible by primary funding from the National Institutes of Health (NIH), 9000 Rockville Pike, Bethesda, Maryland 20892 under award 1R01HD089850-01 for the implementation of Adolescent transition to adult care for HIV-infected adolescents in Kenya (ATTACH) study. The funders did not play any role in study design, data collection and analysis, decision to publish, or preparation of the manuscript.

**Competing interests:** The authors have declared that no competing interests exist.

## Conclusion

The ATP can be feasibly implemented in HIV care clinics at a modest increase in overall clinic visit cost. Our cost estimates can be used to inform economic evaluations or budgetary planning of adolescent HIV care interventions in Kenya.

## Introduction

Globally, an estimated 1.7 million children under 15 years old are living with HIV, of whom approximately 90% reside in sub-Saharan Africa [1]. Despite widespread availability of antiretroviral therapy (ART) which improves survival of individuals with HIV, pediatric ART coverage has lagged adult coverage with only 47% of children on ART [2–5]. Consequently, while 67% of adults achieved viral suppression in 2020, only 40% of children and adolescent were suppressed [1]. Poor transition from pediatric to adult HIV treatment programs and low rates of HIV-status disclosure from guardians to adolescents with HIV contribute to suboptimal ART continuation [6, 7]. Evidence suggests that retention 1–2 years post-transition to adult care ranges from 37% to 95% depending on the setting, with fewer data from low-and-middle income countries [8]. Studies from African countries show that between 60% to 80% of adolescents prefer to remain in pediatric care [9, 10], and ART adherence among those who transition may be approximately 90% during the first year, but rapidly declines over time [11]. Interventions are needed to ensure adequate transition to adult HIV care [12–15].

Given the urgent need to develop tools that aid health care workers (HCWs) to facilitate effective transition of children and adolescents to adult care, the Adolescent Transition To Adult Care for Adolescents Living with HIV in Kenya (ATTACH) study developed an adolescent transition package (ATP) [16], which incorporates evidence-based interventions to improve the readiness of young people (15 to 24 years) with HIV to transition into HIV adult care. ATP effectiveness was evaluated in a cluster randomized clinical trial (RCT; Trial Registration Number NCT03574129) across 20 clinics in four Kenyan counties, assigned 1:1 to either the ATP intervention or standard of care for one year. Results showed the scores of participants in ATP clinics were significantly higher for overall readiness (adjusted mean difference 1.7, 95%CI 0.3, 3.1) and HIV literacy (adjusted mean difference 1, 95%CI 0.2, 1.7) than those in the standard of care clinics [17].

We sought to estimate the total cost of an HIV care visit for adolescent transition, and the incremental cost of implementing the ATP package for care transition in Kenya. These costs can provide important evidence about the economic feasibility of the ATP intervention and inform policymakers in making resource allocation decisions. Our results can also inform future economic evaluations of HIV interventions.

## Methods

We conducted a time-driven, activity-based, micro-costing from the provider perspective to estimate the total cost of an adolescent HIV clinic visit and the incremental costs of implementing the ATP [18]. We categorized each service into activities (e.g., patient's overall health assessment, booklet review, viral load testing), and each activity into inputs (e.g., time of physician to conduct the health assessment, the booklet itself, laboratory materials).

## Study setting

This costing study was conducted within the ATTACH study, a hybrid type 1 effectiveness-implementation cluster RCT located in Nairobi, Homabay, Kajiado and Nakuru counties in Kenya [16]. The trial aimed to evaluate the impact of ATP on youth readiness for transition from pediatric to adult HIV care. Overall, 20 HIV care clinics were randomized 1-to-1 to implement either the ATP or continue standard of care HIV care transition visits for adolescents and young adults aged 10–24 living with HIV (ALH) [17]. The standard of care practices for adolescent transition and disclosure followed national guidelines and focused on HIV literacy and importance on medication adherence [19].

## ATP intervention

The ATP toolkit consists of two evidence-based interventions [16, 20]. The first is a structured booklet, 'Taking Charge,' used to guide the discussion about transitioning to adult care. Booklet design was informed by the US-based 'Got Transition' and USAID tools [21, 22] and adapted to the Kenyan setting in collaboration with stakeholders (young people, policy makers, community members, caregivers, and clinicians). The second intervention was the validated Namibia HIV Disclosure Intervention, a comic book 'Why I take my medicine', that uses a narrative about soldiers fighting a battle with an infection as a rationale to take ART [23]. In intervention sites, ATP was provided during regularly scheduled clinic visits, every three months for one year. All clinics utilized a readiness assessment tool to track individual transition readiness scores [16]. Healthcare providers at intervention clinics received training on ATP toolkit and continuous support on ATP implementatioan during the first 6 months of the study. Nurses most frequently administered the ATP intervention, followed by counsellors and peer educators [17]. All three cadre of HCWs are involved in adolescent care and were trained to deliver ATP [16].

## Data collection

We conducted time and motion observation at a subset of study clinics. Clinic visits were divided into activities and we enumerated inputs needed to complete each activity and then assigned prices to each input [24]. Data collection was conducted in-person by 10 trained research assistants using REDCap in 13 clinics [25, 26]. We created a flow-map (Fig 1) of the activities that comprise a clinic visit with the ATP intervention. Clinic visit activities were assumed to be the same in intervention and control clinics except for "Review Booklet", which only occurred in intervention clinics.

Research assistants utilized a time and motion instrument to measure how long HCWs spent in each clinic activity. In addition, we conducted costing interviews with HCWs to assess their routine clinical duties, perceived time spent on activities, and resources used. HCWs were also asked to estimate the number of patients seen per day. HCWs estimated the average duration of ordinary non-transition ART visit (for treatment continuation) to measure personnel time. We measured resources and utilization using both the time and motion observations and HCW interviews.

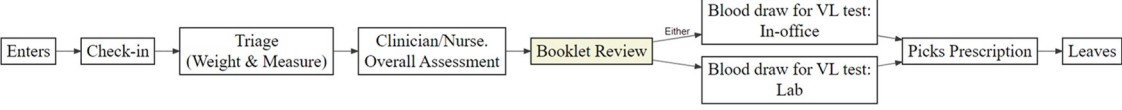

**Fig 1. Flow map of HIV transition services for adolescents.**

We used the ATTACH project expense reports to obtain the unit cost of resources utilized. When data on unit cost was not available we searched for Kenya-specific information on websites offering medical supplies; if this information was not available we consulted with in country experts (program officers and research coordinators). We calculated the cost of developing and delivering the booklet, office supplies, and medical equipment. We estimated the unit cost of the ATP booklet by adding up costs used in development (content creation, illustration, and translation) and printing (delivery to the facility), divided by the average number of ALH seen per day, as reported by HCWs. Wages of physicians, nurses, and counselors were collected from administrative offices from participating facilities. Research costs were excluded to estimate the programmatic cost of ATP implementation.

Resources were categorized into human resources (personnel wages), capital resources (equipment), or disposable (materials used only once) [27]. For human resources, we collected monthly salaries and estimated the cost per minute of HCW time. Capital resources were annualized assuming 2 years of useful life for medical and information technology (e.g., tablets, phones) [28]. To estimate the cost per session of using the ATP Booklet, we utilized the ratio of the total development cost and the average annual client volume across sites, assuming 3 years of useful life. Costs were collected in 2021 Kenyan Shillings (Ksh) and converted to United States Dollars (USD) using an exchange rate of 107.8 Ksh per USD [29].

## Analysis

We used the time and motion data to determine if the activities in control and intervention sites were the same (apart from the 'booklet review' which occurred in the intervention sites only). We estimated the average and standard deviation of the time spent in each activity and resources utilized. From the costing interviews, we determined the resources used as reported by HCWs. We assessed the differences in time spent in each activity measured by the time and motion and costing interviews using the $I^2$ statistic [30] with a threshold of equality of 60% [31]. If the parameter was below this threshold, for any activity (indicating low heterogeneity across information sources) we used pooled estimates. Otherwise, we used the time and motion estimate as it reflected direct observation. We also conducted a qualitative assessment of the concordance of the resources used by activity between instruments. Further, to account for the possibility of shared activities across HCWs, we pooled the data from both instruments to identify the most likely HCW (i.e., physician, nurse, counselor) conducting each activity, which enabled accurate wage estimation for personnel time spent per activity. We used Sankey plots to visualize the distribution of activities across HCW types and differences between control and intervention facilities.

We estimated the cost per activity as the sum of the amount spent on a resource and its price. We calculated the cost per visit as the summation of the cost of each activity separately for the control and intervention clinic visits and conducted a probabilistic sensitivity analysis to estimate the average cost per session and uncertainty intervals [32]. We used a Monte Carlo simulation to obtain 1,000 deviates from the prior distribution of each costing parameter using a random number generator (See S1 Appendix for details). We used the cost per session and reported volume of clients per day to estimate the annual cost of providing ART continuation care for a site under standard of care and using the ATP toolkit.

We also estimated the cost of a pediatric visit for ART-continuation (in which the provider does not offer counseling to transition to adult HIV care). This estimate allowed us to quantify the additional cost associated with transition to adult care services in both control and intervention facilities. We obtained an overall estimate of the time spent in ART continuation visits from the HCWs interviews. Additionally, we conducted a scenario analysis assuming 5 years of useful life for the ATP booklets, with the volume of clients kept constant.

### Ethical approvals

The University of Washington Institutional Review Board and the Kenyatta National Hospital Ethics and Research Committee approved this sub-study of ATTACH. HCWs provided oral informed consent to participate in the study.

## Results

### Sample description

Data was collected from March 11 to April 30, 2021, in 13 clinics: 7 intervention and 6 standard of care clinics. S2 Appendix provides details on each clinic. HCW's (N = 37) demographic characteristics were similar in control and intervention facilities (Table 1). Most (21/37) HCWs identified their role in the clinics as 'other.' We found qualitative differences in cadre composition and activity reported across intervention and control sites A greater proportion of HCWs listed their role as counselor in control compared to intervention sites (42.1% vs 18.5%, respectively).Nurse counselor was more frequently listed in intervention clinics (63% vs 23% in control, respectively), as was clinical officers (15.8% vs. 7.4% in control, respectively). See S3 Appendix for more details. We found less variability among reported activities across sites. Prescription dispensing was more frequently listed in control clinics (14.6% in control and 7.5% in intervention), while triage was similar across both sites (18.8% in control and 15% in intervention, respectively), as was counseling (27.1% in control and 26.2% in intervention,

**Table 1. Respondents' demographic and labor characteristics by site type.**

| | Control (N = 17) | Intervention (N = 20) | Total (N = 37) | p value |
|---|---|---|---|---|
| Gender | | | | 0.509[1] |
| Female | 13 (76.5%) | 17 (85.0%) | 30 (81.1%) | |
| Male | 4 (23.5%) | 3 (15.0%) | 7 (18.9%) | |
| Education | | | | 0.395[1] |
| Polytechnic | 0 (0.0%) | 2 (10.0%) | 2 (5.4%) | |
| Primary | 2 (11.8%) | 1 (5.0%) | 3 (8.1%) | |
| Secondary | 4 (23.5%) | 7 (35.0%) | 11 (29.7%) | |
| University/college | 11 (64.7%) | 10 (50.0%) | 21 (56.8%) | |
| Years at Clinic | | | | 0.759[2] |
| Mean | 4.59 | 4.25 | 4.41 | |
| SD | 2.43 | 3.91 | 3.27 | |
| Years providing HIV care Total | | | | 0.184[2] |
| Mean | 6.41 | 8.25 | 7.41 | |
| SD | 2.87 | 4.92 | 4.16 | |
| Years providing HIV care C&A | | | | 0.803[2] |
| Mean | 5.94 | 5.65 | 5.78 | |
| SD | 3.11 | 3.80 | 3.46 | |
| Patient seen per day Total | | | | 0.320[2] |
| Mean | 29.24 | 24.00 | 26.41 | |
| SD | 15.34 | 16.04 | 15.73 | |
| Patients seen per day, C&A HIV+ | | | | 0.311[2] |
| Mean | 6.59 | 9.65 | 8.24 | |
| SD | 3.55 | 11.82 | 9.04 | |

Notes: [1]Person's Chi-squared test.

[2]Analysis of variance (ANOVA) test. C&A: Children and Adolescent. HIV+: Diagnosed with HIV.

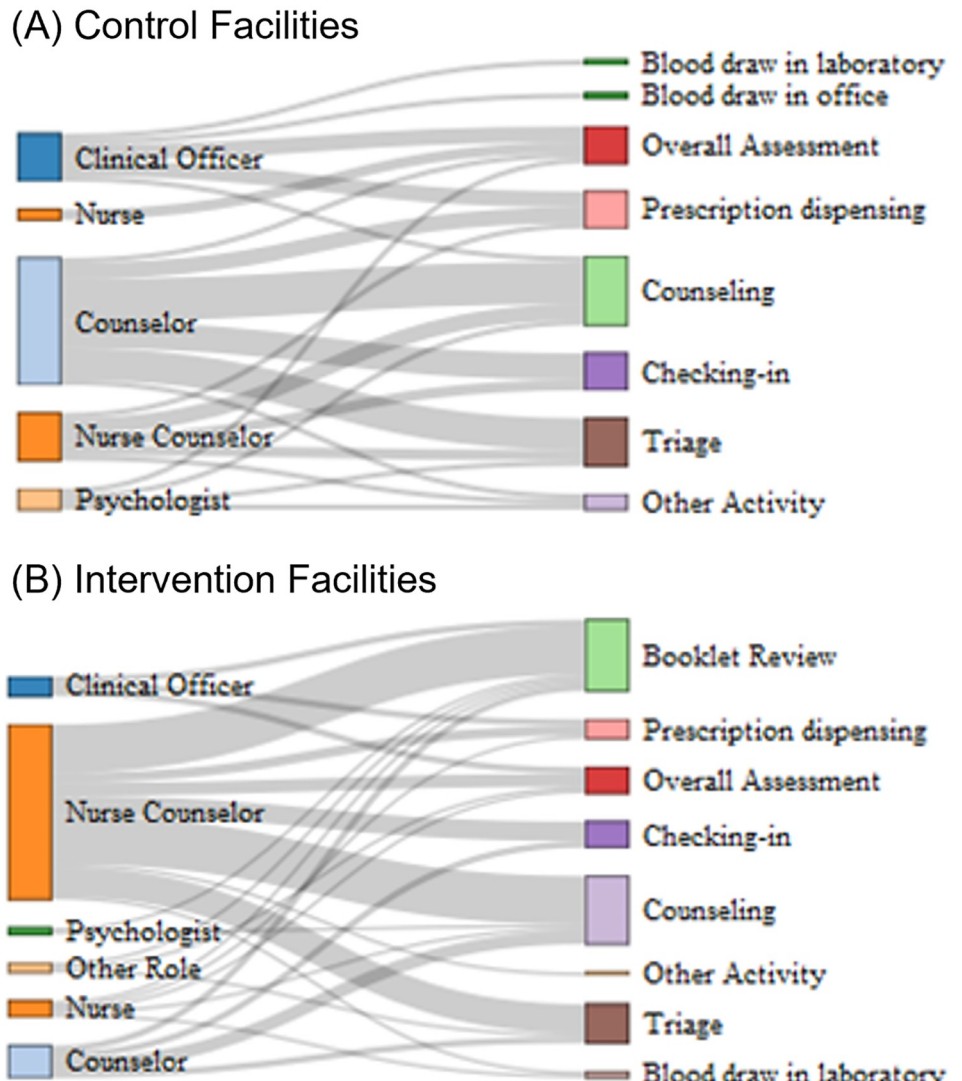

**Fig 2.** Distribution of activities by role in control (A) and intervention (B) facilities.

respectively). See S4 Appendix for more details. Fig 2 shows the Sankey plots of activities by role for intervention sites. In control sites, counseling was mostly performed by counselors followed by nurse counselors. The opposite was found in intervention sites; nurse counselors performed most of the counseling, triage, and booklet review. In both, clinical officer most frequently performed prescription dispensing, overall assessment, and blood draws.

### Estimated time spent per activity

We collected time and motion observations for all activities aside from the blood draw, which was estimated from the costing interviews. We generated pooled estimates for the six activities where enough information was collected by both instruments. S5 and S6 Appendices provide a detailed description of the estimated times spent in each activity by site type and data collection instrument. We observed low heterogeneity across instruments by activity ($I^2$ 0–41%). Reviewing the ATP booklet took 13.6 minutes (98%CI 10.3, 16.8), counseling took 19.1

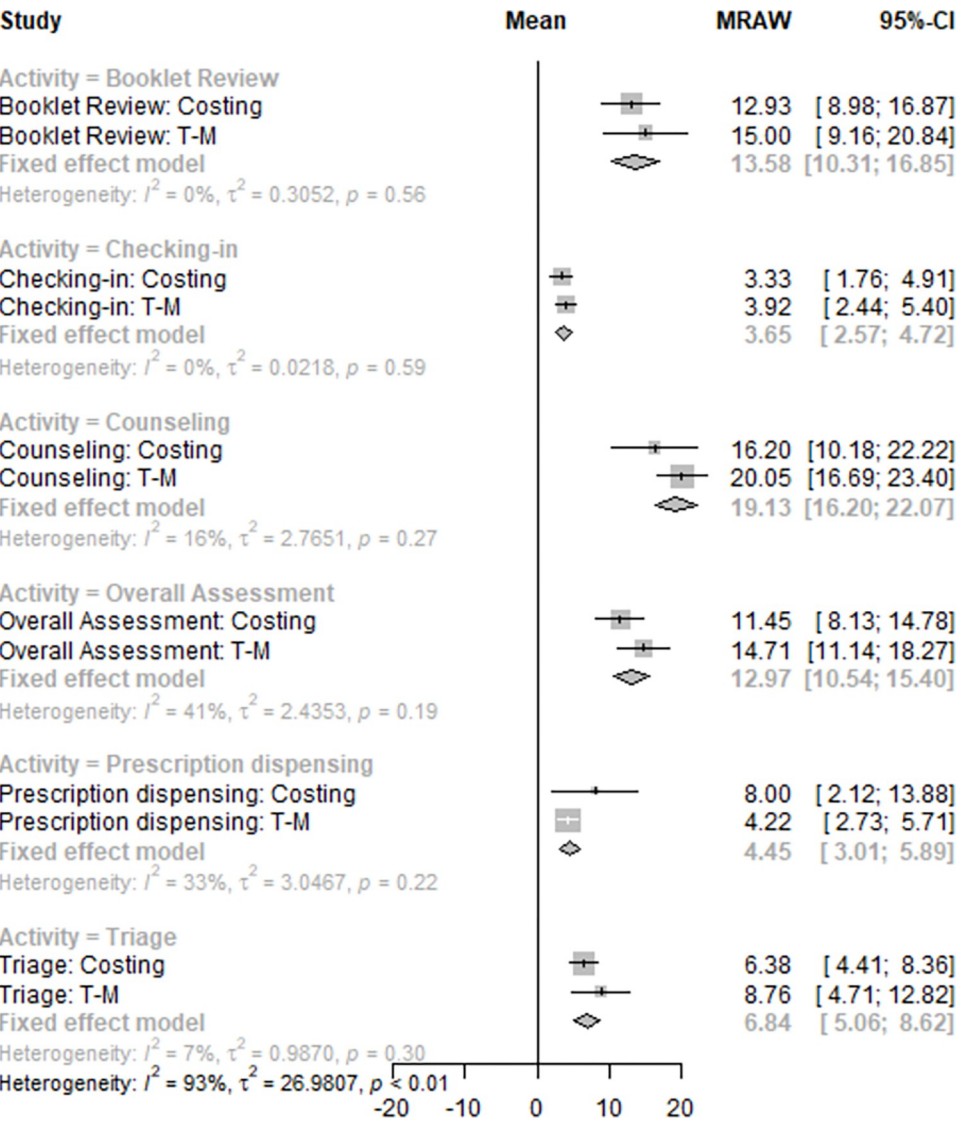

**Fig 3. Estimated time spent in each activity from time and motion and costing instruments.**

minutes (95%CI 16.2, 22.1), and overall assessment took 12.9 minutes (95%CI 10.5, 15.4) (Fig 3). We estimated that the entire clinic visit took 52.1 minutes in the standard of care, and 65.6 minutes in the intervention. After removal of the transition related components, we estimated that an ART-continuation-only visit took 34.7 minutes, including all activities from checking-in to prescription dispensing; 12.7 minutes (SD 3.78min) were dedicated to the clinic consultation.

## Resource utilization per activity

We collected price data for most resources used; items estimated using expert consultation and internet searches played a minor role in overall cost. See S7 Appendix for details on resources utilized per activity and S8 Appendix for a detailed list of items, price, and information source. HCWs reported seeing 6.6 and 9.7 ALH daily on average, in control and intervention sites, respectively, for an overall average of 8.2; we found no evidence of statistically

**Table 2. Cost per ART transition session in control and intervention facilities, broken down by category of cost.**

| Category | ATP clinic visit | | Standard of care clinic visit | | Difference |
|---|---|---|---|---|---|
| | Mean Costs (USD) | 95%CI | Mean cost (USD) | 95%CI | Mean (USD) |
| ATP Booklet | 1.2 | 1.2–1.2 | NA | NA | 1.2 |
| Clinical Equipment | 0.0 | 0.0–0.0 | 0.0 | 0.0–0.0 | - |
| Communication equipment[1] | 0.1 | 0.1–0.1 | 0.1 | 0.1–0.1 | - |
| Disposable Materials | 6.7 | 6.2–7.6 | 6.4 | 6.0–7.0 | 0.2 |
| Educational tools | 9.5 | 9.5–9.5 | 9.5 | 9.5–9.5 | - |
| Human Resources | 6.0 | 4.6–9.3 | 4.6 | 3.4–7.8 | 1.4 |
| Information System[2] | 0.4 | 0.2–0.7 | 0.4 | 0.2–0.7 | - |
| Office Supplies | 9.0 | 7.5–10.7 | 8.7 | 7.3–10.5 | 0.3 |
| Total | 32.9 | 30.5–36.8 | 29.8 | 27.5–33.4 | 3.1 |

Notes: [1]Includes electronic devices used for appointment management (i.e., scheduling, reminder, cancellation).

[2] Includes Electronic Medical Records system. See S8 Appendix for details on all resources included in each category.

significant differences (p-value of the t-test for difference in means 0.31; Table 1). The total cost of developing the ATP booklet was 11,114 USD, and the unitary price per use of an ATP Booklet was 1.2 USD, accounting for average client volume and assuming a useful life of 3 years.

## Cost estimates

The estimated cost per individual visit of ART transition care in control facilities was 29.8USD (95%CI 27.5, 33.4) and 32.9USD (95%CI 30.5, 36.8) in facilities delivering the ATP intervention. The main cost drivers in both intervention and control sites were educational tools (9.5USD in both settings), office supplies (9.0USD and 8.7USD), disposable materials (6.7USD and 6.4USD), and human resources (6.0USD and 4.6USD). (Table 2). Assuming 3 clinic visits per year, the annual cost of the intervention per client was 98.7USD. ART-continuation-only visits (i.e., without transition to adult care components) cost 19.5USD (95%CI 17.5, 24) per session.

The incremental cost of providing ART transition care using the ATP was 3.1USD (95% CI 3.0, 3.4), calculated as the difference in cost per visit between intervention and control sites. This difference was almost entirely explained by the booklet review with a cost of 2.8USD per session, consisting of the booklet itself and HCW time (Table 3). The use of the ATP booklet cost per session of 1.2USD, while cost in human resources was 1.4USD higher in intervention sites (6USD per session) than in control (4.6USD per session) (Table 2). Fig 4 shows the total and component costs per session for intervention and control facilities. Using the reported client volume, we estimated the annual cost of providing ALH HIV-care at 71,679USD for an average control clinic, and 115,882USD. The incremental cost associated with the use of the ATP is 44,202USD per year and clinic.

Fig 5 shows costs density plots for the intervention and control clinics. We observed no overlap between the plots, implying that the difference between the estimates for intervention and control settings holds under parameter uncertainty of the resources employed to provide care.

In the scenario analysis assuming 5 years of useful life for the ATP booklets, we found a reduction in costs of 0.5USD in the booklet review; the total costs in the intervention clinics were reduced to 32.4USD (95%CI 30, 36.3).

**Table 3. Cost per ART transition session in control and intervention facilities, broken down by activity.**

| Activity | ATP clinic visit | | Standard of care clinic visit | | Difference |
|---|---|---|---|---|---|
| | Mean Cost (USD) | 95%CI | Mean Cost (USD) | 95%CI | Mean (USD) |
| Blood draw in laboratory | 1.9 | 1.5–2.8 | 1.0 | 0.8–1.4 | 0.3 |
| Blood draw in office | NA | NA | 0.6 | 0.5–0.8 | |
| Booking | 0.9 | 0.0–4.1 | 0.9 | 0.0–4.1 | - |
| Booklet Review | 2.8 | 2.5–3.2 | NA | NA | 2.8 |
| Checking-in | 1.6 | 1.4–1.8 | 1.6 | 1.4–1.8 | - |
| Counseling | 8.7 | 7.3–10.3 | 8.7 | 7.3–10.3 | - |
| Overall Assessment | 14.4 | 13.9–15.0 | 14.4 | 13.9–15.0 | - |
| Prescription dispensing | 1.8 | 1.6–2.1 | 1.8 | 1.6–2.1 | - |
| Triage | 0.8 | 0.6–1.0 | 0.8 | 0.6–1.0 | - |
| Total | 32.9 | 30.5–36.8 | 29.8 | 27.5–33.4 | 3.1 |

Notes: See S8 Appendix for details on all resources included in each activity.

## Discussion

In this activity-based micro-costing, we estimate that the average cost of a standard of care visit to provide transition support to ALH in Kenya was 29.8USD, incorporating the ATP intervention resulted in an estimated cost of 32.9USD per visit. Thus, adding the ATP implementation to improve the transition to adult care for adolescents costs an additional 3.1USD per visit. Additionally, the cost of a pediatric ART-continuation-only visit was approximately 19.5USD. We found no differences in time needed for activities with reported data in both the

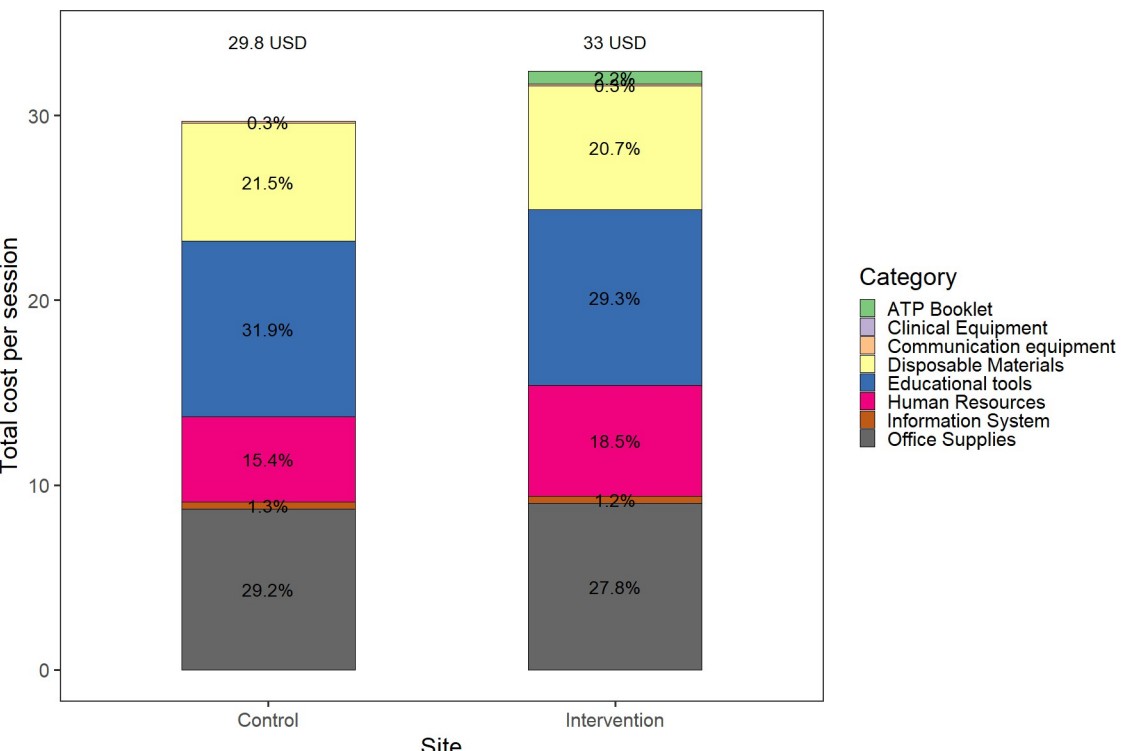

**Fig 4. Proportion attribute to each cost category in intervention and control sites.**

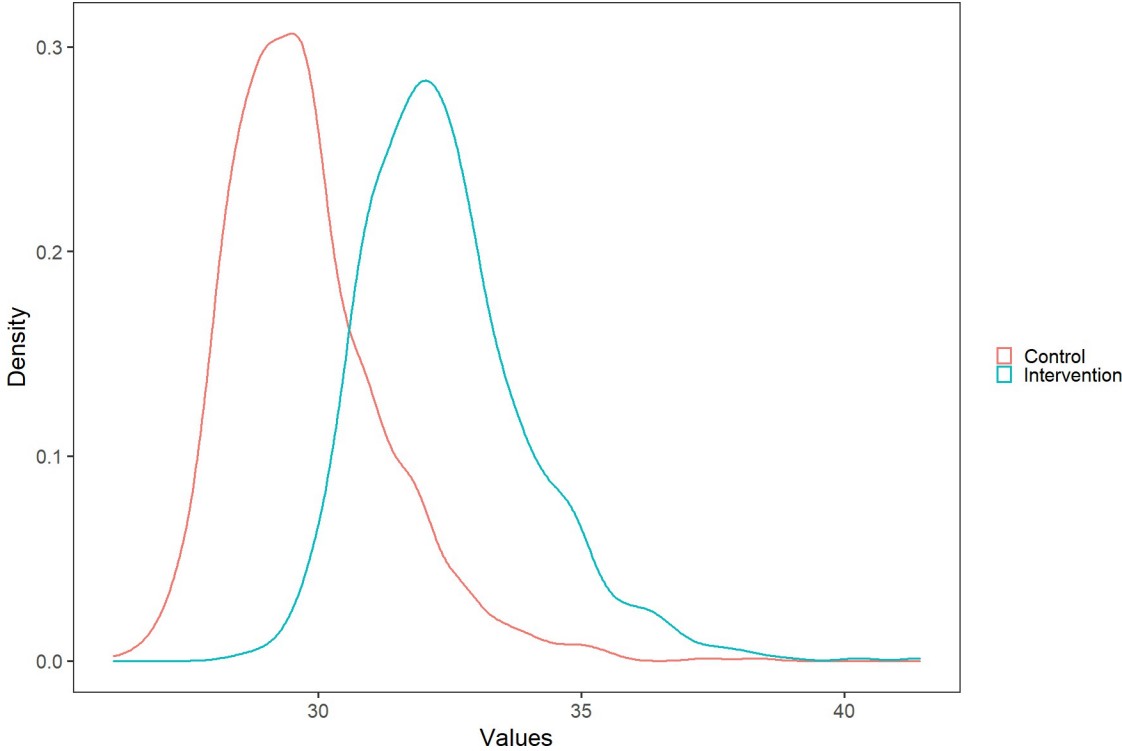

**Fig 5. Density plot of the cost of care in control and intervention clinics.**

time and motion observations and costing interviews, providing evidence of internal consistency and suggesting that both methods can yield equivalent results in the same setting.

Across costs categories, the largest driver of the ATP intervention cost was the ATP booklet, which generated a difference of 2.8USD (90.3% of the total difference) between control and intervention sites. The estimated cost of the ATP booklet was dependent upon the reported volume of ALH at the clinic. While we found some differences across control and intervention sites, these were not significant and therefore we opted for the average across all to enhance the accuracy of the metric. Increasing the booklet useful life to 5 years led to a slight reduction of 0.5USD in the total cost per session in intervention sites. This implies that the main cost driver is the time spent by HCWs on the delivery of the booklet. The only other difference observed was the with blood-draw practices across sites (Table 3). In the context of a programmatic scale up of the ATP intervention, the booklet design and delivery are the most important additional cost.

Our cost estimates for clinic visits are higher than those reported in the literature. The Institute for Health Metrics and Evaluation [33] found that the cost per ART visit for the general population in Kenya was 21USD in a national or provincial hospital, 18USD in a district or private hospital, and 8USD in a public health center. Larson, et al [34] found that the cost per person retained in ART for at least 12 months was 16.3USD per clinic visit in three rural clinics in Kenya. These costs were updated to 2021 USD using the Kenyan inflation index [35]. Our results may be higher since we include the cost of providing counseling and guidance for transition to adult care in addition to regular services. Additionally, while other studies analyzed the general population, we focused on pediatric patients that may require more HCW time, particularly for counseling and delivery of health-related information. The CDC and the Kenyan Ministry of Health [36] estimated the annual cost of a pediatric patient in $126.7 and

Dutta, et al [37] estimated the overall annual cost of ART services in Kenya, including drug costs, for patients aged 0 to 2 years in 103.7USD, for patients aged 2 to 14 years 114.4USD, and for patients aged 15 to 19, 139.8USD. However, without information on frequency of visits, these estimates are not directly comparable to our findings.

This study has several limitations. First, it was not possible to find empiric cost data for all resources. However, the resources for which we used alternative sources represent a small proportion of the total cost. Second, we had limited sample size for each resource used to provide the service. We aimed to account for these two sources of uncertainty in the probabilistic sensitivity analysis. Third, we conducted micro-costing in one region of Kenya, which may not generalize to other areas of the country. Furthermore, costs were collected alongside a research trial which may not be representative of routine clinical practice. However, our time and motion observations included only costs and HCW time for routine clinic visits and not research-related activities. Fourth, this analysis was conducted from a provider perspective so does not account for societal costs including those incurred by patients. Finally, while we find that the incremental cost of adding the ATP intervention to a clinic visit is small, the costs increase with higher clinic volume and may be substantial. However, improving adolescent retention on ART may result in averted costs of HIV-related hospitalizations. Future analyses are needed to evaluate affordability and cost-effectiveness. Additionally, the estimated economic costs include those associated with designing the intervention and may not be applicable to clinics that incorporate the tool as its currently designed.

The strengths of our study include using rigorous methodology and triangulation of multiple data sources for activity identification, data collection, and statistical analysis. At each step of the data collection, we assessed the level of agreement of our instruments and corroborated their accuracy with field-based information. Further, we conducted a comprehensive statistical analysis including an uncertainty analysis to describe the level of confidence in our estimates.

## Conclusions

We find that the adolescent HIV care transition with the ATP intervention can be implemented at reasonable cost at HIV clinic visits in Kenya. Our cost estimates provide empiric data for future analyses evaluating the cost-effectiveness and budget impact of implementing adolescent transition to adult care in Kenya and similar settings.

## Supporting information

**S1 Appendix. Monte Carlo simulation used to estimate parameter uncertainty.**
(DOCX)

**S2 Appendix. Number of records collected on each instrument, by clinic.**
(DOCX)

**S3 Appendix. Respondents' self-reported roles and activities by type of facility.**
(DOCX)

**S4 Appendix. Respondents' reported activities by type of facility.**
(DOCX)

**S5 Appendix. Time-motion instrument: Estimated times per activity.**
(DOCX)

**S6 Appendix. Costing instrument: Estimated times per activity.**
(DOCX)

**S7 Appendix. Resources utilization by activity.**
(DOCX)

**S8 Appendix. Unitary price for each resource included in the analysis.**
(DOCX)

**S1 Questionnaire.**
(DOCX)

## Author Contributions

**Conceptualization:** Enrique M. Saldarriaga, Irene Njuguna, Grace John-Stewart, Monisha Sharma.

**Data curation:** Enrique M. Saldarriaga, Alvin Onyango.

**Formal analysis:** Enrique M. Saldarriaga.

**Funding acquisition:** Dalton Wamalwa, Grace John-Stewart.

**Investigation:** Enrique M. Saldarriaga.

**Methodology:** Enrique M. Saldarriaga, Kristin Beima-Sofie, Irene Njuguna, Monisha Sharma.

**Resources:** Cyrus Mugo, Alvin Onyango, Monisha Sharma.

**Supervision:** Monisha Sharma.

**Writing – original draft:** Enrique M. Saldarriaga.

**Writing – review & editing:** Kristin Beima-Sofie, Dalton Wamalwa, Cyrus Mugo, Irene Njuguna, Alvin Onyango, Grace John-Stewart, Monisha Sharma.

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
