## [Decision Letter · Decision Letter 0]

14 Jul 2023

PONE-D-22-28780Estimating the costs of adolescent HIV care visits and an intervention to facilitate transition to adult care in KenyaPLOS ONE

Dear Dr. Saldarriaga,

Thank you for submitting your manuscript to PLOS ONE. After careful consideration, we feel that it has merit but does not fully meet PLOS ONE’s publication criteria as it currently stands. Therefore, we invite you to submit a revised version of the manuscript that addresses the points raised during the review process.

ACADEMIC EDITOR:

We have had difficult securing a second reviewer for your manuscript. To prevent any further delay in rendering a decision on your manuscript, I have decided to base this decision on reviewer #1 and my own review of your manuscript. Your paper was of interest and can provide a value to the existing literature.  As you can see below, there were both strengths and weaknesses in your paper. There are important areas of your paper that require careful attention. From my own reading of your paper, I am in agreement with the reviewer that your paper will make an important contribution and that your paper will benefit from a thoughtful revision. Please revise your paper in accordance with the reviews. In addition to the comments by reviewer #1, particularly regarding the language/grammar, Tables/Figures, and methodology, please address the following comments:

Methods:

1.    Clarify the main study outcomes from the original clinical trial and this study. Page 4 says the main study outcomes are transition readiness, retention in care, and viral suppression. The authors should clarify that this was for the parent study. 

2.    A description of the ATP would help readers better understand the intervention. Also a brief description of the training, timing and delivery of the ATP would assist in the interpretation. Also clarify if the ATP is delivered individually or to a group of adolescents. 

Results:

3.    Results – resources utilization per activity. The first 3 sentences should be in the methods not results section.

4.    Would report the estimated cost of the ATP per clinic per adolescent not by day (8.24 adolescents per day?) since this number can be quite variable. 

5.    Should also include the total cost of the ATP per adolescent per year. How many sessions per year is the ATP?

6.    What is the total cost of the ATP per adolescent? For example how much does it cost for an adolescent to complete the package and transition to adult care? 

Please be sure to include a cover letter with your revision that addresses all of the reviews and comments as well as my comments, point-by-point. Your revision should also carefully follow the journal style as presented in the Instructions for Authors available at the journal website.  

We look forward to receiving your revised manuscript.

Kind regards,

Brian C. Zanoni, MD

Academic Editor

PLOS ONE

Journal Requirements:

2. Please ensure you have included the registration number for the clinical trial referenced in the manuscript.

3. Please include a complete copy of PLOS’ questionnaire on inclusivity in global research in your revised manuscript. Our policy for research in this area aims to improve transparency in the reporting of research performed outside of researchers’ own country or community. The policy applies to researchers who have travelled to a different country to conduct research, research with Indigenous populations or their lands, and research on cultural artefacts. The questionnaire can also be requested at the journal’s discretion for any other submissions, even if these conditions are not met.  Please find more information on the policy and a link to download a blank copy of the questionnaire here: https://journals.plos.org/plosone/s/best-practices-in-research-reporting. Please upload a completed version of your questionnaire as Supporting Information when you resubmit your manuscript.

Additional Editor Comments:

We have had difficult securing a second reviewer for your manuscript. To prevent any further delay in rendering a decision on your manuscript, I have decided to base this decision on reviewer #1 and my own review of your manuscript. Your paper was of interest and can provide a value to the existing literature. As you can see below, there were both strengths and weaknesses in your paper. There are important areas of your paper that require careful attention. From my own reading of your paper, I am in agreement with the reviewer that your paper will make an important contribution and that your paper will benefit from a thoughtful revision. Please revise your paper in accordance with the reviews. In addition to the comments by reviewer #1, particularly regarding the language/grammar, Tables/Figures, and methodology, please address the following comments:

Methods:

1. Clarify the main study outcomes from the original clinical trial and this study. Page 4 says the main study outcomes are transition readiness, retention in care, and viral suppression. The authors should clarify that this was for the parent study.

2. A description of the ATP would help readers better understand the intervention. Also a brief description of the training, timing and delivery of the ATP would assist in the interpretation. Also clarify if the ATP is delivered individually or to a group of adolescents.

Results:

3. Results – resources utilization per activity. The first 3 sentences should be in the methods not results section.

4. Would report the estimated cost of the ATP per clinic per adolescent not by day (8.24 adolescents per day?) since this number can be quite variable.

5. Should also include the total cost of the ATP per adolescent per year. How many sessions per year is the ATP?

6. What is the total cost of the ATP per adolescent? For example how much does it cost for an adolescent to complete the package and transition to adult care?

Please be sure to include a cover letter with your revision that addresses all of the reviews and comments as well as my comments, point-by-point. Your revision should also carefully follow the journal style as presented in the Instructions for Authors available at the journal website.

Reviewers' comments:

Reviewer's Responses to Questions

**Comments to the Author**

1. Is the manuscript technically sound, and do the data support the conclusions?

Reviewer #1: Yes

2. Has the statistical analysis been performed appropriately and rigorously? 

Reviewer #1: I Don't Know

3. Have the authors made all data underlying the findings in their manuscript fully available?

Reviewer #1: Yes

4. Is the manuscript presented in an intelligible fashion and written in standard English?

Reviewer #1: Yes

5. Review Comments to the Author

Reviewer #1: This is a timely discussion about transitioning from adolescent to adult models of HIV care using a US-based tool adapted for Kenyan adolescents. The study represents a low cost, less time intensive intervention that has the potential to be useful in many areas (when adapted appropriately); however, this study is currently limited to the need for additional discussion about the tool, further details about the settings that it has been trialed in, and reassessing grammar.

METHODS: Overall, more information is needed. Please provide an overview of the ATP model. While there is a citation of the study methods previously cited, a brief description of ATP should be presented here as well. Also what types of providers are usually in these clinics. Figure 2 (which needs a title) provides inferences to this briefly, but isn't mentioned until the results section. Also, please described the difference between nurses, nurse counselors, and providers. Within the "Study Setting" there is mention of a "Review Booklet." What is this and who normally does this activity. The first sentence after Figure 1 should read as follows: "We developed a time and motion instrument to quantify how long HCWs SPENT..." There is also mention of "ATTACH budgetary." Please describe what this is and provide a citation for it. Also provide a rationale for why the ATP costs were calculated in the manner that they were. The entire ANALYSIS section should be reviewed for grammar.

RESULTS: What is "member of Operation Triple Zero." Grammer should be checked within the section as well. In the subsection titled "Resource utilization per activity" (please note, I have removed the "S" after "Resources."), provide additional information about how internet searches for marker prices were obtained.

TABLES: Table 1 needs a more descriptive title. In Table 2- please provide a footnote describing communication equipment. (This table would be easier to understand with a description of the ATP intervention in the methods section.)

FIGURE: All figures need better resolution and most are blurry and the lightest gray font is challenging to read if someone prints the manuscript in black and white. All figures in this section need a title. Figure 2 (which I'm assuming is the one with "Control Facilities" as A and "Intervention facilities" as B but is currently unlabeled including no Figure number and no title) would benefit from a footnote box describing these roles. In Figure 4- there are several numbers in the smaller boxes within either of the 2 columns that need to be displayed differently as they overlap with other number.

6. PLOS authors have the option to publish the peer review history of their article (what does this mean?). If published, this will include your full peer review and any attached files.

Reviewer #1: No

---

## [Author Response · Author response to Decision Letter 0]

8 Oct 2023

(For a better formatted document, please see the attached filed "Respond to Comments")

Responses:

The authors thank the efforts made by the Editor to guarantee an appropriate review of our manuscript. Please find our answers in italics to all comments and requests for changes. We are also submitting the revised version in track changes and clean versions.

Editor

We have had difficult securing a second reviewer for your manuscript. To prevent any further delay in rendering a decision on your manuscript, I have decided to base this decision on reviewer #1 and my own review of your manuscript. Your paper was of interest and can provide a value to the existing literature. As you can see below, there were both strengths and weaknesses in your paper. There are important areas of your paper that require careful attention. From my own reading of your paper, I am in agreement with the reviewer that your paper will make an important contribution and that your paper will benefit from a thoughtful revision. Please revise your paper in accordance with the reviews. In addition to the comments by reviewer #1, particularly regarding the language/grammar, Tables/Figures, and methodology, please address the following comments:

Response: We thank the Editor for providing feedback to facilitate the processing of our manuscript.

Methods:

1. Clarify the main study outcomes from the original clinical trial and this study. Page 4 says the main study outcomes are transition readiness, retention in care, and viral suppression. The authors should clarify that this was for the parent study.

Response: We have clarified this in the second paragraph of the Introduction, where we describe the findings from the parent trial as part of providing context for our costing analysis. In addition, we have added a sub-section called “ATP Intervention” in the Methods section, where we provide much more detail on the ATP intervention procedures.

Page 4 and 5,

“ATP Intervention

The ATP toolkit consisted of two evidence-based interventions[16,20]. The first is a structured booklet, ‘Taking Charge,’ used to guide the discussion about transitioning to adult care. Booklet design was informed by the US-based ‘Got Transition’ and USAID tools[21,22] and adapted to the Kenyan setting in collaboration with stakeholders (young people, policy makers, community members, caregivers, and clinicians). The second intervention was the validated Namibia HIV Disclosure Intervention, a comic book ‘Why I take my medicine’, that uses a narrative about soldiers fighting a battle with an infection as a rationale to take ART[23]. In intervention sites, ATP was provided during regularly scheduled clinic visits, every three months for one year. All clinics utilized a readiness assessment tool to track individual transition readiness scores[16].

Healthcare providers at intervention clinics received training on ATP toolkit use and were provided continuous support on APT implementation during the first 6 months of the study. Nurses most frequently administered the ATP intervention, followed by counsellors and peer educators[17]. While all three cadre of HCWs are involved in adolescent care and were trained to deliver ATP[16]. Further, nurses have a broader scope of work while nurse counselors are more specialized in providing coping tools and guidance[24].”

2. A description of the ATP would help readers better understand the intervention. Also a brief description of the training, timing and delivery of the ATP would assist in the interpretation. Also clarify if the ATP is delivered individually or to a group of adolescents.

Response: We agree. We have added the “ATP Intervention” in the Methods section for this purpose.

Results:

3. Results – resources utilization per activity. The first 3 sentences should be in the methods not results section.

Response: We agree with the editor. We have substantially revised the Methods and Results section to better separate the information in each section as well as to improve clarity, grammar, and phrasing.

4. Would report the estimated cost of the ATP per clinic per adolescent not by day (8.24 adolescents per day?) since this number can be quite variable.

Response: We apologize for the confusion. We present the estimated annual costs per clinic of providing ART transition care using the ATP toolkit, for which we used the clinic client volume. We have rewritten the ‘Cost estimates’ sub-section to improve clarity. 

Page 11

“Using the reported client volume of 8.24 ALH per day, the annual cost of ART transition care with ATP was 98,963.2USD per year in each clinic.”

5. Should also include the total cost of the ATP per adolescent per year. How many sessions per year is the ATP? 

Response: The intervention lasted for one year at an expected rate of one visit every three months (Njuguna et al, The Lancet HIV, 2022). There are no specific guidelines for the standard of care. We used three visits a year to estimate the annual cost; as detailed in the first paragraph of “Cost estimates.” 

PAGE 5

“In intervention sites, ATP was provided during regularly scheduled clinic visits, every three months for one year. All clinics utilized a readiness assessment tool to track individual transition readiness scores[16].”

Page 10

“Assuming 3 clinics visits per year, the annual cost of the intervention per client was 98.7USD. ART-continuation-only visits cost 19.5USD (95%CI 17.5, 24) per session.”

However, the focus of the study is to present the difference between the cost per session, which could then be used in economic analysis at varying specific frequencies depending on the context.

6. What is the total cost of the ATP per adolescent? For example how much does it cost for an adolescent to complete the package and transition to adult care?

Response: Please see our answer to the previous comment.

 

Reviewer 1

This is a timely discussion about transitioning from adolescent to adult models of HIV care using a US-based tool adapted for Kenyan adolescents. The study represents a low cost, less time intensive intervention that has the potential to be useful in many areas (when adapted appropriately); however, this study is currently limited to the need for additional discussion about the tool, further details about the settings that it has been trialed in, and reassessing grammar.

Response: We thank the reviewer for the time and effort in reviewing our manuscript and positive feedback. We agree with the reviewer’s assessment, and have now revised the paper to reflect the suggested edits.

METHODS: 

Overall, more information is needed. Please provide an overview of the ATP model. While there is a citation of the study methods previously cited, a brief description of ATP should be presented here as well. Also what types of providers are usually in these clinics. Figure 2 (which needs a title) provides inferences to this briefly, but isn't mentioned until the results section. Also, please described the difference between nurses, nurse counselors, and providers.

Response: We agree the intervention should be presented at greater detail. We have added the sub-section “ATP Intervention” to the Methods section. This section also includes our findings from the cluster randomized controlled trial regarding the health care workers who most frequently administered the intervention and a brief description of the differences between types of health care workers.

Pages 4 and 5

“ATP Intervention

The ATP toolkit consisted of two evidence-based interventions[16,20]. The first is a structured booklet, ‘Taking Charge,’ used to guide the discussion about transitioning to adult care. Booklet design was informed by the US-based ‘Got Transition’ and USAID tools[21,22] and adapted to the Kenyan setting in collaboration with stakeholders (young people, policy makers, community members, caregivers, and clinicians). The second intervention was the validated Namibia HIV Disclosure Intervention, a comic book ‘Why I take my medicine’, that uses a narrative about soldiers fighting a battle with an infection as a rationale to take ART[23]. In intervention sites, ATP was provided during regularly scheduled clinic visits, every three months for one year. All clinics utilized a readiness assessment tool to track individual transition readiness scores[16].

Healthcare providers at intervention clinics received training on ATP toolkit use and were provided continuous support on APT implementation during the first 6 months of the study. Nurses most frequently administered the ATP intervention, followed by counsellors and peer educators[17]. While all three cadre of HCWs are involved in adolescent care and were trained to deliver ATP[16]. Further, nurses have a broader scope of work while nurse counselors are more specialized in providing coping tools and guidance[24].”

In addition, we have expanded the information about the ATP in the Introduction.

Pages 3 and 4

“Given the urgent need to develop tools that aid health care workers (HCWs) to facilitate effective transition of children and adolescents to adult care, the Adolescent Transition To Adult Care for Adolescents Living with HIV in Kenya (ATTACH) study (National Institutes of Health, Grant Number 1R01HD089850-01) developed an adolescent transition package (ATP)[16], which incorporates evidence-based interventions to improve the readiness of young people, (15 to 24 years) with HIV to transition into HIV adult care. ATP effectiveness was evaluated in a cluster randomized clinical trial (RCT) across 20 clinics in four Kenyan counties, assigned 1:1 to either the ATP intervention or standard of care for one year. Results showed the scores of participants in ATP clinics were significantly higher for overall readiness (1.7, 95%CI 0.3, 3.1) and HIV literacy (1, 95%CI 0.2, 1.7) than those in the standard of care clinics[17].

We sought to estimate the total cost of an HIV care visit for adolescent transition, and the incremental cost of implementing the ATP package for care transition in Kenya. These costs can provide important evidence about the economic feasibility of the ATP intervention and inform policymakers in making resource allocation decisions. Our results can also inform future economic evaluations of HIV interventions.”

Within the "Study Setting" there is mention of a "Review Booklet." What is this and who normally does this activity. 

Response: We have added information about this in the sub-section “ATP Intervention”, Methods section (page 4 and 5; please see previous answer).

The first sentence after Figure 1 should read as follows: "We developed a time and motion instrument to quantify how long HCWs SPENT..." 

Response: This has been corrected. We thank the reviewer for the careful reading of our manuscript

There is also mention of "ATTACH budgetary." Please describe what this is and provide a citation for it. 

Response: Thank you for pointing this out. We meant to say the “research project expense information.” This data is internal to the execution of the research project and allowed us to obtain estimates of unitary costs of the resources used to provide services. Supplemental Materials 5 contains an exhaustive list of the unitary costs estimated and the source used for each item.

Also provide a rationale for why the ATP costs were calculated in the manner that they were. 

Response: We have now added a paragraph at the beginning of the Methods section providing rationale for the methodology we used.

Page 4

“We conducted a time-driven activity-based, micro-costing analysis from the provider perspective to estimate the total cost of an adolescent clinic visit and the incremental costs of implementing the ATP[18]. We categorized each service into activities (e.g., patient’s overall health assessment, booklet review, viral load testing), and each activity into inputs (e.g., time of physician to conduct the health assessment, the booklet itself, laboratory materials).”

The entire ANALYSIS section should be reviewed for grammar.

Response: We have substantially revised the Analysis section to improve grammar, readability, and overall clarity.

RESULTS: 

What is "member of Operation Triple Zero." 

Response: We initially used this term to provide some examples of how participants described their role. We have removed “operation triple zero” from the list because is less self-explanatory than the other ones and could be distracting for the readers. 

Grammer should be checked within the section as well. 

Response: We have revised the Results section to improve grammar, readability, and overall clarity.

In the subsection titled "Resource utilization per activity" (please note, I have removed the "S" after "Resources."), provide additional information about how internet searches for marker prices were obtained.

Response: We thank the reviewer for this comment. We realize this information is better suited in the Methods sections, under Data Collection. We have moved this information and provided more details to clarify our procedures.

Page 6

“We used the ATTACH project expenses information obtained via field data collection to obtain the unit cost of resources utilized. When data on unit price was not available we searched for Kenya-specific information on websites offering medical supplies; if this information was not available we consulted with in country experts (program officers and research coordinators). We calculated the cost of developing and delivering the booklet, office supplies, and medical equipment. We estimated the unit cost of the ATP booklet by adding up costs used in development (content creation, illustration, and translation) and printing (delivery to the facility), divided by the average number of ALH seen per day, as reported by HCWs. Wages of physicians, nurses, and counselors were collected from administrative offices from participating facilities. Research costs were excluded to estimate the programmatic cost of ATP implementation in a clinic setting.”

TABLES: 

Table 1 needs a more descriptive title. 

Response: Thank you for the suggestion, we have improved the Table’s title.

Page 19

“Table 1. Respondents’ demographic and labor characteristics by site type.”

In Table 2- please provide a footnote describing communication equipment. (This table would be easier to understand with a description of the ATP intervention in the methods section.)

Response: We have expanded the information provided in Table 2 and added a footnote for Table 3 as well.

Page 20

“Notes: 1Includes electronic devices used for appointment management (i.e., scheduling, reminder, cancellation). 2Includes Electronic Medical Records system. See Supplemental material 6 for details on all resources included in each category.”

FIGURE: 

All figures need better resolution and most are blurry and the lightest gray font is challenging to read if someone prints the manuscript in black and white. All figures in this section need a title. 

Response: We apologize for the inconvenience. Per the journals style requirements, we included the figures titles in the manuscript and uploaded the figures separately. However, this time we have included the figures’ title in the ‘description’ during the uploading process, so the caption will hopefully show next to the images in the submission document. 

Regarding the figures’ definition. We agree it could be improved. We are submitting higher quality images.

Figure 2 (which I'm assuming is the one with "Control Facilities" as A and "Intervention facilities" as B but is currently unlabeled including no Figure number and no title) would benefit from a footnote box describing these roles. 

Response: We have added a description of the roles in the sub-section “ATP Intervention”, Methods section.

Page 5

“Healthcare providers at intervention clinics received training on ATP toolkit use and were provided continuous support on APT implementation during the first 6 months of the study. Nurses most frequently administered the ATP intervention, followed by counsellors and peer educators[17]. While all three cadre of HCWs are involved in adolescent care and were trained to deliver ATP[16]. Further, nurses have a broader scope of work while nurse counselors are more specialized in providing coping tools and guidance[24].”

In Figure 4- there are several numbers in the smaller boxes within either of the 2 columns that need to be displayed differently as they overlap with other number.

Response: We agree. The new images also improve readability. However, please consider that the submission document stretches all images to a full page, when they are meant to be just a fraction of it, which affects its quality.

---

## [Decision Letter · Decision Letter 1]

27 Nov 2023

PONE-D-22-28780R1Estimating the costs of adolescent HIV care visits and an intervention to facilitate transition to adult care in KenyaPLOS ONE

Dear Dr. Saldarriaga,

Thank you for submitting your manuscript to PLOS ONE. After careful consideration, we feel that it has merit but does not fully meet PLOS ONE’s publication criteria as it currently stands. Therefore, we invite you to submit a revised version of the manuscript that addresses the points raised during the review process.

Your paper was of interest to the reviewers. As you can see below, the reviewers identified both strengths and weaknesses in your paper. The reviewers noted important areas of your paper that require a minor revision. From my own reading of your paper, I am in agreement with the reviewers that your paper will make an important contribution and that your paper will benefit from a minor revision. Please revise your paper in accordance with the reviews, particularly the comments by reviewer #3. 

Please be sure to include a cover letter with your revision that addresses all of the reviews and comments as well as my comments, point-by-point. Your revision should also carefully follow the journal style as presented in the Instructions for Authors available at the journal website.  

We look forward to receiving your revised manuscript.

Kind regards,

Brian C. Zanoni, MD

Academic Editor

PLOS ONE

Journal Requirements:

Additional Editor Comments (if provided):

Not all of the original reviewers were available to review this re-submission; however, we were able to recruit additional reviewers for your manuscript. Your paper was of interest to the reviewers. As you can see below, the reviewers identified both strengths and weaknesses in your paper. The reviewers noted important areas of your paper that require careful attention. From my own reading of your paper, I am in agreement with the reviewers that your paper will make an important contribution and that your paper will benefit from a minor revision. Please revise your paper in accordance with the reviews paying particular attention to the comments from reviewer #3.

Please be sure to include a cover letter with your revision that addresses all of the reviews and comments point-by-point. Your revision should also carefully follow the journal style as presented in the Instructions for Authors available at the journal website.

Reviewers' comments:

Reviewer's Responses to Questions

**Comments to the Author**

1. If the authors have adequately addressed your comments raised in a previous round of review and you feel that this manuscript is now acceptable for publication, you may indicate that here to bypass the “Comments to the Author” section, enter your conflict of interest statement in the “Confidential to Editor” section, and submit your "Accept" recommendation.

Reviewer #2: All comments have been addressed

Reviewer #3: All comments have been addressed

2. Is the manuscript technically sound, and do the data support the conclusions?

Reviewer #2: Yes

Reviewer #3: Yes

3. Has the statistical analysis been performed appropriately and rigorously? 

Reviewer #2: Yes

Reviewer #3: Yes

4. Have the authors made all data underlying the findings in their manuscript fully available?

Reviewer #2: Yes

Reviewer #3: Yes

5. Is the manuscript presented in an intelligible fashion and written in standard English?

Reviewer #2: Yes

Reviewer #3: Yes

6. Review Comments to the Author

Reviewer #2: The study topic is relevant and the paper is generally well written. The authors have also comprehensively addressed the reviewer comments. I have noted a few issues that should be fixed before publication:

1. The authors should insert a space at the end of sentences and in-text citations; most of the references need this revision.

2. The following sentence in the introduction:

Results showed the scores of participants in ATP clinics were significantly higher

for overall readiness (1.7, 95%CI 0.3, 3.1) and HIV literacy (1, 95%CI 0.2, 1.7) than those in the standard of care clinics[17].

Which measure of association is referenced above? The point estimate and the CI is given, but the measure is not named. Since the measure is documented very well in the referenced study, the authors should consider doing the same here.

3. Methods: in the second paragraph of the ATP intervention section, the sentence below seems incomplete:

While all three cadre of HCWs are involved in adolescent care and were trained to deliver ATP.

4. Results: Resource utilization per per activity section: Is there a reason why t-test is capitalized?

5. Under the costs estimates in the results section: Was there a difference between ART continuation only visits and the usual care delivered to controls? Otherwise, what would explain the difference the cost for the latter (USD 19.5) and the control (USD 29.8)? Therefore, are we comparing the cost of an intervention and usual care (in this case USD 32.9 vs USD 19.5) or two interventions (USD32.9 vs USD29.8)? A clarification here would really be helpful.

Reviewer #3: Thanks for the opportunity to review this revised manuscript. I did not review the original, but have noted the revisions suggested by the previous reviewer and the editor.

This is an interesting manuscript costing the addition of an adolescent transition package (ATP) in Kenya compared to SOC services.

There are a few edits required prior to publication, listed below.

Abstract

- Add that in the randomised clinical trial, facilities were randomised 1:1 to the intervention and control arms

- In results section suggest deleting "with HIV" as this point has already been made

- There appears to be duplication of "healthcare worker time" descriptions which could be combined and summarised

Introduction

- Add reference after 2nd sentence ending "children on ART".

- "...from 37% to 95% depending on THE setting" Suggest adding "the"

- Paragraph 2....delete comma before (15 to 24 years)

Methods

- First sentence suggest a comma after "time-driven"

- In study setting 20 clinics are included however throughout the manuscript and abstract 13 clinics are included, please align

- ATP intervention: Suggest "The ATP toolkit consists of" rather than consisted of

- Typo on page 5 2nd paragraph, 2nd line: APT instead of ATP

- Suggest deleting "while" and starting with "All three cadres of HCW......" add "cadres" instead of "cadre"

- The last sentence "Further, nurses have ......" seems a bit odd, suggest adding more context or deleting.

Results

- There are a number of supplementary materials included, suggest including most relevant as part of the main paper tables/figures or more clearly describing relevant findings since it is a bit distracting to have to flip to supplementary material repeatedly

- There is a large difference in skills for example nurse counsellors at implementation sites compared to controls. Was this purposeful and the reason that these clinics were chosen as implementation sites? This is fine but the context in the results is not clear. Similar with clinical officers. Are these doctors/nurse clinicians/clinical associates? please clarify?

- The difference/significance between lab-based and office based blood draws is not clear to me-please elaborate on the significance of specifying the location of these blood draws. Bloods would need to be done regardless so it's not clear to me why the costs would be different

- Prescription dispensing and triage are important activities but comparing the frequency in the control and intervention clinics also needs context.

- Under cost estimates: 2nd last sentence, delete "s" from 3 clinics..."

- The annual cost of ART transition with ATP of just under 100K USD per clinic seems quite a large cost, even though this translates to and additional 3.1 USD/visit!

Discussion

- 2nd sentence 1st paragraph: suggest adding "...... 3.1 USD extra per visit"

- 2nd paragraph, instead of "volume" of ALH, suggest "number"

- In the discussion there is a lot of focus on the different activities between intervention and control sites, which seems to lack relevance, particularly the discussion between lab- and office based bloods. Please also see previous comment, but the relevance of this and why this should differ between intervention and control is not clear

- As mentioned in the results, the cost of 100K USD per clinic to implement ATP is quite significant and should be discussed. It is expected and affordable within the routine health services?

7. PLOS authors have the option to publish the peer review history of their article (what does this mean?). If published, this will include your full peer review and any attached files.

Reviewer #2: No

Reviewer #3: No

---

## [Author Response · Author response to Decision Letter 1]

11 Dec 2023

Responses:

Reviewer #2: 

The study topic is relevant and the paper is generally well written. The authors have also comprehensively addressed the reviewer comments. I have noted a few issues that should be fixed before publication:

1. The authors should insert a space at the end of sentences and in-text citations; most of the references need this revision.

Response: We have now added an extra space before each of the references in the manuscript.

2. The following sentence in the introduction:

Results showed the scores of participants in ATP clinics were significantly higher

for overall readiness (1.7, 95%CI 0.3, 3.1) and HIV literacy (1, 95%CI 0.2, 1.7) than those in the standard of care clinics[17].

Which measure of association is referenced above? The point estimate and the CI is given, but the measure is not named. Since the measure is documented very well in the referenced study, the authors should consider doing the same here.

Response: Thank you for pointing out this, it was an oversight. We have added the information to the mentioned sentence, it now reads:

“Results showed the scores of participants in ATP clinics were significantly higher for overall readiness (adjusted mean difference 1.7, 95%CI 0.3, 3.1) and HIV literacy (adjusted mean difference 1, 95%CI 0.2, 1.7) than those in the standard of care clinics [17].”

3. Methods: in the second paragraph of the ATP intervention section, the sentence below seems incomplete:

While all three cadre of HCWs are involved in adolescent care and were trained to deliver ATP.

Response: We have now deleted the word “while” from this sentence.

4. Results: Resource utilization per per activity section: Is there a reason why t-test is capitalized?

Response: This was a typo and we have made it lower case. 

5. Under the costs estimates in the results section: Was there a difference between ART continuation only visits and the usual care delivered to controls? Otherwise, what would explain the difference the cost for the latter (USD 19.5) and the control (USD 29.8)? Therefore, are we comparing the cost of an intervention and usual care (in this case USD 32.9 vs USD 19.5) or two interventions (USD32.9 vs USD29.8)? A clarification here would really be helpful.

Response: The main difference is that in both the control and intervention visits, providers offer services for ART-continuation and transition to adult HIV care. In addition, we estimated the cost of a visit focused solely on ART-continuation. We realize this was not sufficiently clear, so we have edited the last paragraph of the sub section “Analysis” from Methods.

Page 7:

“We also estimated the cost of a pediatric visit in which the provider does not offer transition to adult HIV care, that the visit is focused solely on ART-continuation. This estimate allowed us to quantify the additional cost associated with transition to adult care services in both control and intervention facilities. We obtained an overall estimate of the time spent in such visits from the HCWs interviews. Additionally, we conducted a scenario analysis assuming 5 years of useful life for the ATP booklets, with the volume of clients kept constant.”

Reviewer #3: 

Thanks for the opportunity to review this revised manuscript. I did not review the original, but have noted the revisions suggested by the previous reviewer and the editor.

This is an interesting manuscript costing the addition of an adolescent transition package (ATP) in Kenya compared to SOC services.

There are a few edits required prior to publication, listed below.

Abstract

- Add that in the randomised clinical trial, facilities were randomised 1:1 to the intervention and control arms

Response: We have now edited this sentence to read:

“Data was collected from 13 HIV clinics involved in a randomized clinical trial evaluating ATP in western Kenya.” 

We decided not to mention 1:1 randomization as we only collected data in a subset of clinics (and an odd number) which may raise confusion about the methods. Additionally, the focus of the paper is the costing and the protocol has been previously published. Further, the 1:1 randomization is mentioned in the methods section.

- In results section suggest deleting "with HIV" as this point has already been made

Response: We have now deleted “with HIV.”

- There appears to be duplication of "healthcare worker time" descriptions which could be combined and summarized

Response: We have now re-written this section of the abstract.

Introduction

- Add reference after 2nd sentence ending "children on ART".

Response: We have moved the reference to the end of the sentence.

- "...from 37% to 95% depending on THE setting" Suggest adding "the"

Response: We have added “the” to this sentence.

- Paragraph 2....delete comma before (15 to 24 years)

Response: We have deleted this comma.

Methods

- First sentence suggest a comma after "time-driven"

Response: We have added a comma to this sentence. 

- In study setting 20 clinics are included however throughout the manuscript and abstract 13 clinics are included, please align

Response: Although 20 clinics were involved in the main study, we sampled from a subset for the costing study. We now clarify this in the abstract and in the methods section:

Methods, page 5:

We conducted time and motion observation of clinic visits at a subset of study clinics. Clinic visits were divided into activities and we enumerated inputs needed to complete each activity. We assigned prices to each input enumerated [25]. Data collection was conducted in-person by 10 trained research assistants using REDCap in 13 clinics.

Abstract:

Data was collected from 13 HIV clinics involved in a randomized clinical trial evaluating ATP in western Kenya.

- ATP intervention: Suggest "The ATP toolkit consists of" rather than consisted of

Response: We have now made this change.

- Typo on page 5 2nd paragraph, 2nd line: APT instead of ATP

Response: We have corrected this error. Thank you for the careful reading of our manuscript.

- Suggest deleting "while" and starting with "All three cadres of HCW......" add "cadres" instead of "cadre"

Response: We have removed “while” from this sentence.

- The last sentence "Further, nurses have ......" seems a bit odd, suggest adding more context or deleting.

Results

Response: We have deleted this sentence.

- There are a number of supplementary materials included, suggest including most relevant as part of the main paper tables/figures or more clearly describing relevant findings since it is a bit distracting to have to flip to supplementary material repeatedly

Response: We have included the 5 main figures and tables in the manuscript and we present the details of our costing methodology and additional results in the supplemental appendix. As is common with costing papers, there are a large number of supplemental tables. We have elected to leave them in the supplement but have numbered the tables for ease of review.

- There is a large difference in skills for example nurse counsellors at implementation sites compared to controls. Was this purposeful and the reason that these clinics were chosen as implementation sites? This is fine but the context in the results is not clear. Similar with clinical officers. Are these doctors/nurse clinicians/clinical associates? please clarify?

Response: The skills of the personnel were not assessed by our study. The activities each professional performs in every site is independent of both our costing study and the parent trial and did not influence their selection in intervention or control, as this was decided randomly. All cadre of personnel received training to administer the ATP intervention and supervision to ensure quality control. Therefore, we do not think the quality of the intervention or quality of overall ART care would vary across randomly selected intervention and control sites.

- The difference/significance between lab-based and office based blood draws is not clear to me-please elaborate on the significance of specifying the location of these blood draws. Bloods would need to be done regardless so it's not clear to me why the costs would be different

Response: The activity blood draw in laboratory requires the use of additional medical materials which entails costs that are unrelated to the ATP intervention and therefore are only noted as a minor point. Supplemental Table 7 contains a detailed list of the resources used in each activity. Because this is not a main driver in the difference in costs across sites, we have edited the second paragraph of the Discussion accordingly.

Page 12:

“While we found some differences across control and intervention sites, these were not significant and therefore we opted for the average across all to enhance the accuracy of the metric. Increasing the booklet useful life to 5 years led to a slight reduction of 0.5USD in the total cost per session in intervention sites. This implies that the main cost driver is the time spent by HCWs on the delivery of the booklet. The only other difference observed was the with blood-draw practices across sites (Table 3). In the context of a programmatic scale up of the ATP intervention, the booklet design and delivery is the most important additional cost.”

- Prescription dispensing and triage are important activities but comparing the frequency in the control and intervention clinics also needs context.

Response: We did not intend to make inferences about the difference in activity reporting between intervention and control sites as our objective was to collect the most accurate estimates of time and resources spent in each activity. The differences we found are merely qualitative. We have edited the sub-section “Sample description” from Results to clarify.

Page 8:

“We found qualitative differences in cadre composition and activity reported across intervention and control sites. HCW roles varied across control and intervention clinics. A greater proportion of HCWs listed their role as counselor in control compared to intervention sites (42.1% vs 18.5%, respectively).Nurse counselor was more frequently listed in intervention clinics (63% vs 23% in control, respectively), as was clinical officers (15.8% vs. 7.4% in control, respectively). See Supplemental Table 2 for more details. We found less variability among reported activities across sites. Prescription dispensing was more frequently listed in control clinics (14.6% in control and 7.5% in intervention), while triage was similar across both sites (18.8% in control and 15% in intervention, respectively), as was counseling (27.1% in control and 26.2% in intervention, respectively). See Supplemental Table 3 for more details.”

- Under cost estimates: 2nd last sentence, delete "s" from 3 clinics..."

Response: We have now deleted this extra “s”.

- The annual cost of ART transition with ATP of just under 100K USD per clinic seems quite a large cost, even though this translates to and additional 3.1 USD/visit!

Response: We have now addressed this point in the results section to provide more accurate estimates of the incremental cost associated with the intervention. 

Page 8:

“Using the reported client volume, we estimated the annual cost of providing ALH HIV-care at 71,679USD for an average control clinic, and 115,882USD. The incremental cost associated with the use of the ATP is 44,202USD per year and clinic.”

Discussion

- 2nd sentence 1st paragraph: suggest adding "...... 3.1 USD extra per visit"

Response: We have altered that sentence to the following: 

“Thus, adding the ATP implementation to improve the transition to adult care for children and adolescents in Kenya was costs 3.1USD per visit.”

- 2nd paragraph, instead of "volume" of ALH, suggest "number"

Response: We have now changed this sentence. We have elected to say “volume” as it’s common in costing studies to incorporate the clinic volume into the costs of services.

“The estimated cost of the ATP booklet was dependent upon the reported volume of ALH at the clinic.”

- In the discussion there is a lot of focus on the different activities between intervention and control sites, which seems to lack relevance, particularly the discussion between lab- and office based bloods. Please also see previous comment, but the relevance of this and why this should differ between intervention and control is not clear

Response: We added the discussion about blood draw to explain the only other difference in costs beside the booklet, but we do agree it was unnecessary detail. We have edited the second paragraph of the Discussion to address this.

Page 12:

“Across costs categories, the largest driver of the ATP intervention cost was the ATP booklet, which generated a difference of 2.8USD (90.3% of the total difference) between control and intervention sites. The estimated cost of the ATP booklet was dependent upon the reported volume of ALH at the clinic. While we found some differences across control and intervention sites, these were not significant and therefore we opted for the average across all to enhance the accuracy of the metric. Increasing the booklet useful life to 5 years led to a slight reduction of 0.5USD in the total cost per session in intervention sites. This implies that the main cost driver is the time spent by HCWs on the delivery of the booklet. The only other difference found was associated with blood-draw practices across sites. In the context of a programmatic scale up of the ATP intervention, the booklet design and delivery is the most important additional cost.”

- As mentioned in the results, the cost of 100K USD per clinic to implement ATP is quite significant and should be discussed. It is expected and affordable within the routine health services?

Response: Although the economic cost of implementing the ATP intervention may be substantial at clinics with a large volume of ART care transition visits for adolescents, there is likely cost saving to the health system associated with retaining adolescents on ART. The affordability (or budget impact) of this intervention is not within the scope of this study, which does not provide a full economic evaluation. Future studies are needed to assess affordability and cost-effectiveness (value for money). Additionally, we present economic costs of ATP implementation. Now that the intervention has already been developed, these startup costs would likely not be incurred to clinics choosing to implement the transition intervention. 

We have added the following to the discussion section:

Page 14:

“Finally, while we find that the incremental cost of adding the ATP intervention to a clinic visit is small, the costs increase with higher clinic volume and may be substantial. However, improving adolescent retention on ART may result in averted costs of to HIV-related hospitalizations. Future analyses are needed to evaluate affordability and cost-effectiveness. Additionally, the estimated economic costs include those associated with designing the intervention and may not be applicable to clinics that incorporate the tool as its currently designed.”

---

## [Editor Report · Decision Letter 2]

19 Dec 2023

Estimating the costs of adolescent HIV care visits and an intervention to facilitate transition to adult care in Kenya

PONE-D-22-28780R2

Dear Dr. Saldarriaga,

We’re pleased to inform you that your manuscript has been judged scientifically suitable for publication and will be formally accepted for publication once it meets all outstanding technical requirements.

Kind regards,

Brian C. Zanoni, MD

Academic Editor

PLOS ONE

Additional Editor Comments (optional):

The authors have significantly improved their manuscript and adequately responded to the minor revisions indicated by the prior reviewers.
---

## [Editor Report · Acceptance letter]

20 Jan 2024

PONE-D-22-28780R2 

PLOS ONE

Dear Dr. Saldarriaga, 

I'm pleased to inform you that your manuscript has been deemed suitable for publication in PLOS ONE. Congratulations! Your manuscript is now being handed over to our production team.

Kind regards, 

on behalf of

Dr. Brian C. Zanoni 

Academic Editor

PLOS ONE